

# iPro-CSAF: identification of promoters based on convolutional spiking neural networks and spiking attention mechanism

Qian Zhou[1,2,3], Jie Meng[1,2,3] and Hao Luo[4]

[1] Hebei Key Laboratory of Bioelectromagnetics and Neural Engineering, School of Health Sciences and Biomedical Engineering, Hebei University of Technology, Tianjin, China
[2] Tianjin Key Laboratory of Bioelectromagnetic Technology and Intelligent Health, School of Health Sciences and Biomedical Engineering, Tianjin, China
[3] State Key Laboratory of Reliability and Intelligence of Electrical Equipment, Hebei University of Technology, Tianjin, China
[4] Department of Physics, School of Science, Tianjin University, Tianjin, China

Corresponding author
Qian Zhou, qzhou@hebut.edu.cn

## ABSTRACT

A promoter is a DNA segment which plays a key role in regulating gene expression. Accurate identification of promoters is significant for understanding the regulatory mechanisms involved in gene expression and genetic disease treatment. Therefore, it is an urgent challenge to develop computational methods for identifying promoters. Most current methods were designed for promoter recognition on few species and required complex feature extraction methods in order to attain high recognition accuracy. Spiking neural networks have inherent recurrence and use spike-based sparse coding. Therefore, they have good property of processing spatio-temporal information and are well suited for learning sequence information. In this study, iPro-CSAF, a convolutional spiking neural network combined with spiking attention mechanism is designed for promoter recognition. The method extracts promoter features by two parallel branches including spiking attention mechanism and a convolutional spiking layer. The promoter recognition of iPro-CSAF is evaluated by exhaustive promoter recognition experiments including both prokaryotic and eukaryotic promoter recognition from seven species. Our results show that iPro-CSAF outperforms promoter recognition methods which used parallel CNN layers, methods which combined CNNs with capsule networks, attention mechanism, LSTM or BiLSTM, and CNNs-based methods which needed priori biological or text feature extraction, while our method has much fewer network parameters. It indicates that iPro-CSAF is an effective computational method with low complexity and good generalization for promoter recognition.

## INTRODUCTION

A promoter is a non-coding short region (about 100 to 1,000 base pairs) located close to a transcription start site (TSS) in DNA sequences which initiates the transcription of DNA. It has an important role in regulating gene expression. Promoter mutations could lead to disruption of gene expression which contributes to tumor development and rare diseases

(*Melnikova, 2012*). Promoters are critical for controlling the expression of therapeutic genes and minimizing potential safety risks during gene therapy treatments (*Geurts et al., 2007*). Accurate identification of promoters is significant for understanding the regulatory mechanisms involved in gene expression and genetic disease treatments. In prokaryotes, different types of promoters are responsible for expression of different genes. The types of prokaryotic promoters are determined by different σ factors which are responsible for binding to different promoters (*Browning & Busby, 2004*), while in eukaryotes, there are three types of promoters including RNA pol I promoters, RNA pol II promoters and RNA pol III promoters (*Carter & Drouin, 2009*) which are responsible for the transcription of different genes. Compared to eukaryotic promoters, prokaryotic promoters are commonly shorter and have simpler structure. Prokaryotic promoters contain −10 box around −10 bp upstream of TSS and −35 box around −35 bp upstream of TSS. While eukaryotic promoters span a wide range of DNA sequences and contain TATA box located about 25 bp to 35 bp upstream of the TSS, CAAT box and GC box located at −40 bp to −110 bp, CpG island and others. The TATA boxes exist only in eukaryotic promoters. Broadly speaking, eukaryotic promoters can be categorized into TATA and non-TATA promoters based on the presence of TATA box in sequences. These two types of promoters have obviously different structural properties in DNA duplex stability, bendability and curvature (*Yella & Bansal, 2017*).

With the development of high-throughput sequencing technology, the number of biological sequences is exploding. In bioinformatics fields researchers have developed many machine learning methods and especially deep learning methods to help solving problems in many fields including gene expression (*Al taweraqi & King, 2022*), protein structures (*AlQuraishi, 2021*; *Tubiana, Schneidman-Duhovny & Wolfson, 2022*) genomic (*Dalla-Torre et al., 2024*) and single-cell biology (*Cui et al., 2024*), *etc.* These methods achieved state-of-the-art results in the prediction of biological sequences, structures and functions.

Experimental methods for promoter recognition are time-consuming and costly. And there is an increasing need of developing computational methods for promoter recognition. A series of computational methods based on machine learning have been proposed to identify promoters. These methods first utilized mathematical and statistical methods to capture complex features of promoters, such as DNA duplex stability features (*de Avila e Silva et al., 2014*) and pseudo-K-tuple nucleotide composition (PseKNC) (*Liu et al., 2018*; *Xiao et al., 2019*). Then machine learning methods were used for promoter recognition, such as support vector machine (SVM) (*de Avila e Silva et al., 2014*), Random Forest (RF) (*Liu et al., 2018*) and so on.

Deep learning is a kind of data-driven technique that can learn features through multi-layer feature extraction from raw data without relying much on priori knowledge. In recent years, deep learning algorithms have been increasingly used in promoter recognition. pcPromoter-CNN (*Shujaat et al., 2020*) and SAPPHIRE.CNN (*Coppens, Wicke & Lavigne, 2022*) adopted one-hot encoding scheme and multiple convolutional neural network (CNN) layers to identify prokaryotic promoters. *Le et al. (2019)* and *Tahir et al. (2020)* used natural language processing methods to obtain mathematical vectors which were

input to multiple CNN layers for prokaryotic promoter recognition. In order to improve the promoter recognition accuracy of computational methods, researchers proposed various predictors by combining CNNs with other deep learning algorithms including capsule layers (*Zhu et al., 2021*; *Moraes et al., 2022*), long short-term memory (LSTM) network (*Oubounyt et al., 2019*; *Ma et al., 2022*), attention mechanism (*Zhang et al., 2022*), *etc*. On the basis of CNNs, Depicter (*Zhu et al., 2021*) used capsule layers to further extract features. DeeProPre (*Ma et al., 2022*) used bidirectional long short-term memory (BiLSTM) and attention mechanism to extract features. iPromoter-CLA (*Zhang et al., 2022*) used capsule layers, BiLSTM, and attention mechanism to obtain more important sequence features. CyaPromBERT (*Mai, Nguyen & Lee, 2022*) utilized Bidirectional Encoder Representations from Transformers (BERT) to perform pre-training on genomes for promoter recognition. Among the above methods, SAPPHIRE.CNN, Depicter, CapsProm, and DeeProPre were used for multi-species promoter recognition, while the remaining methods performed single-species promoter recognition, and only CapsProm performed the recognition of both prokaryotic and eukaryotic promoters. Although existing methods had exhibited encouraging performance, many promoter recognition methods required very complex feature extraction methods in order to attain high recognition accuracy. Also, most deep learning methods were designed for few species. There is an increasing need for developing promoter recognition method with low complexity and good generalization.

Spiking neural networks (SNNs) are the third generation of artificial neural networks with biological plausibility (*Ghosh-Dastidar & Adeli, 2009*). They are considered to be ideal bio-inspired neuromorphic computing paradigm that well mimic the inherent spike-based and event-driven computation in the brain (*Roy, Jaiswal & Panda, 2019*). Since SNNs have inherent recurrence (*Ponghiran & Roy, 2021*) and use spike-based sparse coding (*Roy, Jaiswal & Panda, 2019*), they have good property of processing spatio-temporal information (*Kheradpisheh et al., 2018*) and are well suited for learning sequence information. They are widely used in many biological sequence classification tasks, such as the classification of motor imagery EEG signals (*Virgilio et al., 2020*), emotion recognition from EEG signals (*Luo et al., 2020*), and the classification of ECG signals (*Feng et al., 2022*) and odor signals (*Vanarse et al., 2020*). *Yan, Zhou & Wong (2022)* used a transfer learning method to train SNNs for emotion classification of EEG signals, achieving a smaller SNN with good accuracy. And the power consumption of SNNs was only 13.8% of the CNNs-based EEG emotion recognition scheme, which demonstrated the advantage of SNNs' low power consumption. *Zhou, Qi & Ren (2021)* applied SNNs with reward-modulated spike-timing-dependent plasticity (R-STDP) learning mechanism to make gene essentiality prediction, and they obtained an accuracy of 81.79% for intra-organism predictions, which was significantly better than SVM classifier.

Self-attention mechanism (*Vaswani et al., 2017*) has been successfully applied to natural language processing, computer vision, bioinformatics and so on. The mechanism allows deep learning networks to focus on the connections of separated positions in long-distance sequences. Spiking self-attention (SSA) mechanism (*Zhou et al., 2022*) was specially designed for SNNs, which captured long-distance dependence features and represented

them by spiking sequences. The attention maps are calculated by sparse spike-form Query, Key, and Value. The computation progress of spiking self-attention is efficient and the energy consumption is low. *Abdullah Alohali et al. (2024)* adopted spiking transformer based on SSA for speech enhancement, which improved the ability to learn and process noisy speech. These studies have demonstrated the ability of spiking neural networks for learning sequence information.

Considering the suitability of SNNs for learning sequence information and their advantages of low power consumption, in this article we design a SNN method called iPro-CSAF for promoter recognition by combining convolutional spiking neural networks (CSNNs) with multi-head spiking self-attention mechanism. And we use two parallel branches, of which one is a convolutional spiking layer to extract high-dimensional local features and the other is multi-head SSA mechanism to extract long-distance dependence features from DNA sequences, respectively. The two kinds of features are then fused and input to a spiking fully connected layer to make decisions. The performance of iPro-CSAF is validated on different promoter datasets from seven species which include *Escherichia coli* (*E. coli*), *Bacillus subtilis* (*B. subtilis*), cyanobacteria, *Homo sapiens* (*H. sapiens*), *Drosophila melanogaster* (*D. melanogaster*), *Mus musculus* (*M. musculus*) and *Arabidopsis thaliana* (*A. thaliana*). We make ablation experiments and compare the performance of iPro-CSAF with baseline methods and state-of-art deep learning promoter recognition methods. We also calculate the power consumption of iPro-CSAF. Our main contributions are as follows: (i) We propose a spiking-based promoter recognition model named iPro-CSAF which combines convolutional spiking neural networks with multi-head spiking self-attention mechanism in a parallel structure to extract the spatio-temporal features of promoter sequences. (ii) The recognition performance of iPro-CSAF is evaluated by exhaustive promoter recognition experiments including both prokaryotic and eukaryotic promoter recognition from seven species, and is compared with state-of-art deep learning promoter recognition methods. (iii) We demonstrate that iPro-CSAF is an effective computational method with low complexity and good generalization for both prokaryotic and eukaryotic promoter recognition.

## MATERIALS AND METHODS

### Experimental datasets

The first step in building a useful promoter predictor is to select reliable benchmark and independent datasets for method training and performance validation. Our datasets are from seven species including both prokaryotes and eukaryotes. Prokaryotes include *E. coli*, *B. subtili*, and cyanobacteria. The length of each sample is 81 bp. Eukaryotes include *H. sapiens*, *D. melanogaster*, *M. musculus*, *A. thaliana*. The length of each sample is 300 bp.

For *E. coli* promoter recognition, all the samples of our benchmark dataset are obtained from the dataset constructed by *Xiao et al. (2019)* (https://ars.els-cdn.com/content/image/1-s2.0-S088875431830613X-mmc1.pdf). The independent dataset for *E. coli* promoter recognition is obtained from the dataset constructed by *Liu et al. (2018)* (https://academic.oup.com/bioinformatics/article/34/1/33/4158035#supplementary-data). It contains 2,860 promoter sequences and 2,860 non-promoter sequences. For *B. subtilis* promoter

recognition, all the samples are from the datasets constructed by (*Umarov & Solovyev, 2017*) (https://github.com/solovictor/CNNPromoterData.git). For cyanobacteria promoter recognition, all the samples are obtained from the DB1 datasets constructed by (*Yang et al., 2024*) (https://github.com/Passion4ever/SiamProm).

For promoter recognition of *A. thaliana, M. musculus, D. melanogaster, H. sapiens*, all their samples are obtained from the datasets constructed by *Zhu et al. (2021)* (https://github.com/zhuyaner/Depicter/). The four eukaryotic promoter datasets mentioned above are randomly divided into benchmark datasets and independent datasets at a ratio of 9:1. For each of the eukaryotic promoter datasets, we perform three recognition tasks including TATA promoter and non-promoter recognition, non-TATA promoter and non-promoter recognition, and also TATA&non-TATA promoter and non-promoter recognition. For the benchmark datasets of all these seven species, we validate our method using five-fold cross-validation. After that, in order to prevent experimental over-fitting and further evaluate prediction performance of our iPro-CSAF, independent datasets are used to test the method. The samples of benchmark datasets are summarized in Table 1.

## The overall architecture of iPro-CSAF

iPro-CSAF consists of one-hot coding layer, a spiking feature extraction module and a spiking classification module, as shown in Fig. 1. Spiking feature extraction module consists of a 1-d convolutional spiking layer, a spiking feature fusion module which contains two parallel branches and one concatenation module.

In Fig. 1, the first convolutional spiking layer receives one-hot encoding matrices of promoter and non-promoter samples. And after the spiking neuron layer and batch normalization (BN) layer, samples are converted into discrete binary spikes (*Xiong et al., 2021*), thus realizing the transmission of spikes between different layers. Therefore, the first convolutional spiking layer not only extracts features but also encodes one-hot vectors into spikes. In feature fusion module, each branch further extracts features from the output of first convolutional layer. The output of these two parallel branches is concatenated to achieve the fusion of local and global features of promoter sequences. The spiking features after fusion are fed to a spiking fully connected layer for making decisions. Surrogate gradient (*Neftci, Mostafa & Zenke, 2019*) is used to enable end-to-end training of iPro-CSAF. Dropout operation is adopted in both the feature extractor and the spiking classifier, which can randomly remove some neurons to avoid overfitting (*Dahl, Sainath & Hinton, 2013*).

## One-hot encoding of input promoter sequences

There are four types of nucleotides in a DNA sequence: A (adenine), C (cytosine), G (guanine), and T (thymine). DNA sequences must be converted into numerical vectors to perform feature extraction and classification. The one-hot encoding approach is widely used in the recognition of promoter sequences in deep learning. By one-hot encoding, each nucleotide is encoded as a four-dimensional binary vector whose elements represent four types of nucleotides, respectively. One position of the one-hot vector is 1 and the elements of the remaining positions are all 0. The detailed form of each nucleotide encoded by

**Table 1 The statistical summary of promoter and non-promoter benchmark datasets in this study.**

| Species | Recognition tasks | Benchmark datasets | |
|---|---|---|---|
| | | Positive | Negative |
| *E. coli* | Promoter and non-promoter | 3,382 | 3,382 |
| *B. subtilis* | Promoter and non-promoter | 373 | 1,000 |
| cyanobacteria | Promoter and non-promoter | 12,566 | 12,566 |
| *A. thaliana* | TATA promoter and non-promoter | 5,691 | 5,691 |
| | Non-TATA promoter and non-promoter | 14,272 | 14,272 |
| | TATA&non-TATA promoter and non-promoter | 19,963 | 19,963 |
| *M. musculus* | TATA promoter and non-promoter | 2,769 | 2,769 |
| | Non-TATA promoter and non-promoter | 18,936 | 18,936 |
| | TATA&non-TATA promoter and non-promoter | 21,705 | 21,705 |
| *D. melanogaster* | TATA promoter and non-promoter | 2,326 | 2,326 |
| | Non-TATA promoter and non-promoter | 12,631 | 12,631 |
| | TATA&non-TATA promoter and non-promoter | 14,957 | 14,957 |
| *H. sapiens* | TATA promoter and non-promoter | 2,634 | 2,634 |
| | Non-TATA promoter and non-promoter | 22,914 | 22,914 |
| | TATA&non-TATA promoter and non-promoter | 25,548 | 25,548 |

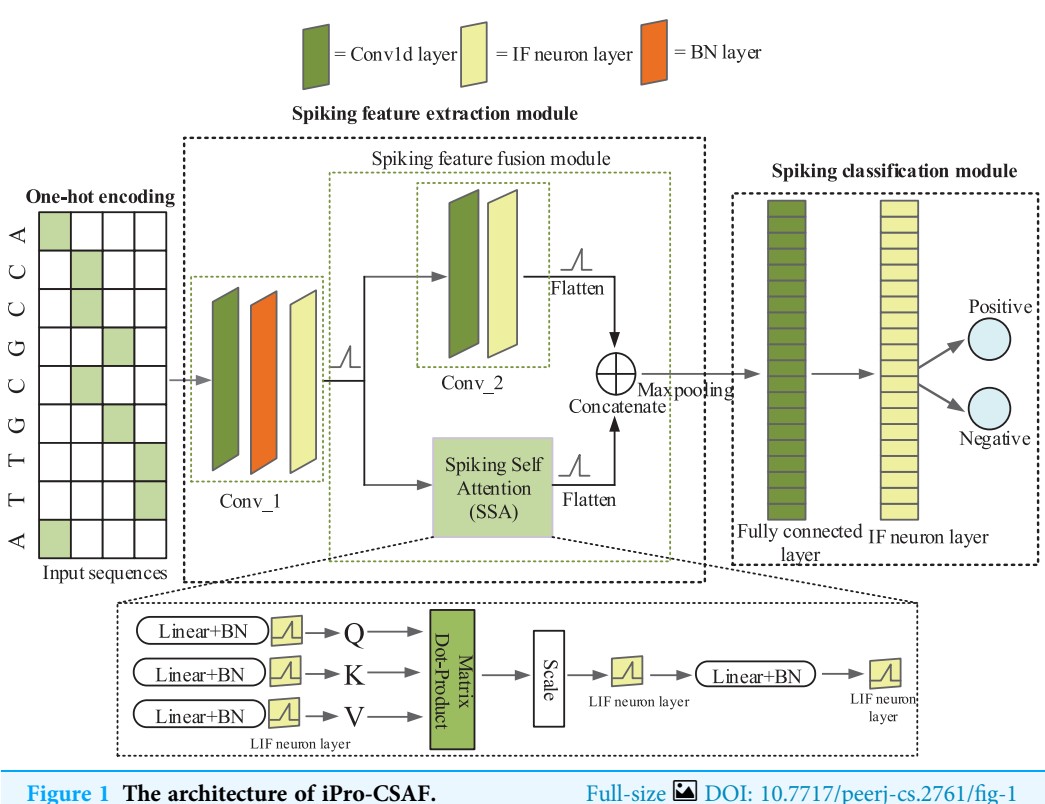

**Figure 1 The architecture of iPro-CSAF.**

one-hot encoding is shown below: $A$ (1, 0, 0, 0), $C$ (0, 1, 0, 0), $G$ (0, 0, 1, 0), $T$ (0, 0, 0, 1). After one-hot encoding, a promoter sequence of length L is encoded into an L × 4 two-dimensional matrix.

## Spiking neuron model

Spiking neuron models simulate the firing of biological neurons and are the basic processing units of SNNs. Frequently used spiking neuron models are the IF model, leaky integration and firing (LIF) model, Hodgkin-Huxley (H-H) model and Izhikevich model. A spiking neuron receives input which changes its neuronal membrane potential, and emits a spike when its membrane potential increases to a prespecified threshold potential. The process of charging, discharging, and resetting of a spiking neuron can be represented by the following three equations:

$$H(t) = f(V(t-1),\ X(t)) \tag{1}$$

$$S(t) = \theta(H(t) - V_{thr}) = \begin{cases} 1,\ H(t) - V_{thr} \geq 0 \\ 0,\ H(t) - V_{thr} < 0 \end{cases} \tag{2}$$

$$V(t) = H(t) * (1 - S(t)) + V_{reset} * S(t) \tag{3}$$

where Eq. (1) describes the charging process of a neuron. $t$ represents simulation time step. $V(t-1)$ is the membrane potential at time step $t-1$. $X(t)$ is the external input. $f(V(t-1),\ X(t))$ is a function describing the updating of neuron membrane potential. For different spiking neuron models, this function is different. $H(t)$ is the hidden state of a neuron, which is instantaneous voltage before the neuron fires a spike at time step $t$. Eq. (2) describes the discharging process of a neuron. $V_{thr}$ refers to the threshold membrane potential of a neuron. If $H(t)$ exceeds the threshold membrane potential, the neuron is activated and emits a spike. Eq. (3) describes the resetting of a neuron's membrane potential. $V_{reset}$ is the reset potential after the neuron sends out a spike. If the neuron does not send out a spike, then $V(t) = H(t)$. If the neuron sends out a spike, then $V(t) = V_{reset}$.

In our convolutional spiking layers and fully connected spiking layer, we use IF neuron model because it has a simple structure and is suitable for large-scale simulation. The membrane potential $V(t)$ remains constant when there is no external input, and the update process of $V(t)$ of IF neuron is shown as follows:

$$dV(t)/dt = R \cdot I(t) \tag{4}$$

where $R$ is the leakage resistance of an IF neuron model. $I(t)$ is the external input current.

In our spiking self-attention module, LIF neuron model is adopted to generate spikes. The update process of $V(t)$ is shown as follows:

$$\tau \cdot dV(t)/dt = -(V(t) - V_{reset}) + R \cdot I(t) \tag{5}$$

where $\tau$ is time constant of membrane potential.

## The convolutional spiking layers

In our method we use one-dimensional convolutional layers which are suitable for processing sequence data. The first convolutional spiking layer receives one-hot encoding

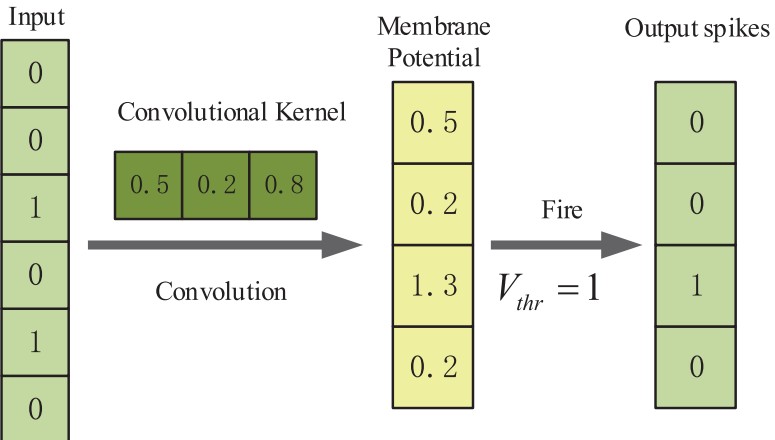

**Figure 2 An example illustration of the spiking convolutional operation based on IF neurons.**

binary matrices, and the remaining convolutional spiking layer receives spikes which are output from the IF neuron layer. An example illustration of the spiking convolutional operation based on IF neurons is shown in Fig. 2. Here, the input is [0, 0, 1, 0, 0, 1, 0]. The size of the one-dimensional convolution kernel is 3 and the stride size is 1. The convolution kernel covers the corresponding positions of the input spiking sequence in turn and performs the convolution operation. After that the feature response map [0.5, 0.2, 1.3, 0.2] is obtained and it represents the membrane potentials of IF spiking neurons in spiking neuron layer. The membrane potential threshold is set to be 1. If the feature value exceeds the membrane potential threshold, the neuron will discharge and release a spike and the membrane potential will be reset. If the feature value is less than the membrane potential threshold, the membrane potential will remain constant. Finally, the output spike train is [0, 0, 1, 0].

## Multi-head spiking self-attention

The input of multi-head spiking self-attention is spike feature maps X from the first convolutional spiking layer, which have the data format T × N × D. Here T is the time step, N is the number of spiking sequences, and D is the dimension of spiking sequences.

The SSA module consists of linear layer, batch normalization layer and spiking neuron layer.

The input is mapped to Query (Q), Key (K), Value (V) by linear layer: Q = X · W_Q, K = X · W_k, V = X · W_v. The weights of linear layer are $W_Q$, $W_K$, $W_V$. Then, Q, K and V are encoded into spike matrices $Q$, $K$, $V$ by being input to batch normalization layer and spiking neuron layer. The calculation equation is as follows:

$$Q = \text{LIF}(\text{BN}(X \cdot W_Q)), K = \text{LIF}(\text{BN}(X \cdot W_K)), V = \text{LIF}(\text{BN}(X \cdot W_V)). \tag{6}$$

Then the attention maps SSA(Q, K, A) are calculated and encoded into spikes by linear layer and spiking neuron layer. The specific process is shown as follows:

$$\text{SSA}(Q, K, A) = \text{LIF}\big(\text{BN}\big(\text{Linear}\big(\text{LIF}\big(QK^{\mathrm{T}}V \cdot s\big)\big)\big)\big) \tag{7}$$

where Q, K, V $\in R^{\mathrm{T} \times \mathrm{N} \times \mathrm{D}}$, and s is a scaling factor to control the large value of an attention map. Multi-head spiking self-attention mechanism (MSSA) is multiple parallel SSA operations. Q, K, V are split into H parts. We run SSA operations for H times. The calculation process of multi-head attention maps are as follows:

$$\text{MSSA}(Q,\ K,\ V) = [\text{SSA}_1(Q_1,\ K_1,\ V_1),\ \text{SSA}_2(Q_2,\ K_2,\ V_2),\ \ldots\ldots\ldots,\ \text{SSA}_H(Q_H,\ K_H,\ V_H)] \tag{8}$$

The attention maps of SSA are obtained by logical AND operation and addition of spike-form Q, K, V. It avoids float-form multiplications.

## Spiking feature fusion module

We propose a spiking feature fusion module which contains two parallel branches and one concatenation module. The two parallel branches are a convolutional spiking layer and a multi-head spiking self-attention module, respectively. Each branch further extracts features from the output of the first convolutional layer. And the concatenation module fuses local features output from the convolutional spiking layer and global features output from the spiking attention mechanism. All features are fused and further transmitted in the form of spikes. They are input to the fully connected spiking layer to make decisions. The feature maps from the convolutional spiking layer and spiking attention mechanism are $E_{conv} \in R^{T \times N \times D}$ and $E_{MSSA} \in R^{T \times N \times D}$, respectively, where T is the time step, N is the number of channels of the feature maps, and D is the dimension of spiking sequences. They are concatenated after being flattened. The feature fusion maps are $E_{concatenation} \in R^{T \times (N \times D + N \times D)}$.

## The surrogate gradient algorithm

Because in the spiking neuron model, the step function $\theta(x)$ in Eq. (2) is non-differentiable (*Guo & Wang, 2023*), it is not possible to train a spiking neural network directly using backpropagation algorithm. We use surrogate gradient learning to train iPro-CSAF. $\theta(x)$ is approximated by a differentiable gating function, known as surrogate gradient. In the forward propagation process, the network still uses the step function $\theta(x)$. In the backpropagation process, the surrogate gradient is calculated by surrogate function $g(x)$. In convolutional layers, we use arctan function $g(x) = (1/\pi) \cdot \arctan((\pi/2) \cdot \alpha \cdot x) + 1/2$. In spiking self-attention, we use sigmoid function $g(x) = sigmoid(\alpha x) = 1/(1 + e^{-\alpha x})$. Here the parameter $\alpha$ is proportional to the degree of similarity between $g(x)$ and $\theta(x)$. It can control the degree of smoothing of the surrogate function. The larger $\alpha$ is, the more prone to vanishing gradient and exploding gradient, which makes the network more difficult to be trained. In arctan function, we set $\alpha = 2$. In sigmoid function, we set $\alpha = 4$.

## Experimental setup and parameter selection

Our network is implemented by utilizing the spiking neural network framework SpikingJelly based on Pytorch and python3.9. The mean-square error (MSE) is used as loss function and the Adam optimization algorithm is adopted.

**Table 2 Hyperparameter selection range.**

| Parameters | Range |
|---|---|
| Time steps | 9 |
| Number of convolution kernels | 64 |
| Size of the convolution kernel | 5 |
| Membrane potential threshold | 1 mV |
| Learning rate | 0.0007, 0.0009 |
| Dropout ratio | 0.3, 0.4, 0.5, 0.6, 0.7 |
| Max pooling layer size and stride size | 2, 2 |
| The number of attention heads | 8 |

The choice of parameters has great influence on promoter recognition performance of computational methods. We perform hyperparameter tuning for the convolutional layers, time steps, learning rate, the number of heads in MSSA and dropout ratio. The range of tuning parameters of our method is shown in Table 2. The values of hyperparameters in Table 2 are slightly different for specific promoter recognition tasks in this study.

The spatial complexity of the model can be represented by the number of model parameters which is often used to measure the performance of deep learning models in terms of complexity. Table 3 gives the specific network structure and parameter numbers of each layer in our method. In the Results & discussion section, we will compare the number of parameters of iPro-CSAF with those of compared methods.

## Method evaluation metrics

Four commonly used metrics are used to assess the performance of the method, including sensitivity (Sn), specificity (Sp), accuracy (Acc), Matthews correlation coefficient (MCC), and F1-score. The metrics are described as follows:

$$recall, \ Sn = TP/(TP + FN) \tag{9}$$

$$Sp = TN/(TN + FP) \tag{10}$$

$$Pre = TP/(TP + FP) \tag{11}$$

$$Acc = (TP + TN)/(TP + TN + FP + FN) \tag{12}$$

$$MCC = (TP \times TN - FP \times FN)/\sqrt{(TP + FP)(TP + FN)(TN + FP)(TN + FN)} \tag{13}$$

$$F1 \ score = 2 \times (Pre \times Sn)/(Pre + Sn) \tag{14}$$

where TP, TN, FP and FN are the number of samples that the method predicts as true positive, true negative, false positive and false negative, respectively. Sp and Sn denote the ability of the method to correctly identify negative and positive samples, respectively. Pre denotes the ratio of true promoters that are recognized as promoters. Acc denotes the percentage of samples correctly categorized by the method. MCC shows the recognition results of methods on unbalanced datasets. F1-score (F1) is a comprehensively evaluation metric which considers precision and recall. The values of sensitivity, specificity, F1 score

**Table 3 Parameter numbers of each layer of iPro-CSAF.**

| Name of layers | Input shape | Hyperparameter | Output shape | Parameter numbers |
|---|---|---|---|---|
| Conv1d | [None, 4, 300] | kernel size = 5<br>Stride = 1 | [None, 64, 300] | 1,280 |
| BatchNorm1d | [None, 64, 300] | – | [None, 64, 300] | 128 |
| MultiStepIFNode | [None, 64, 300] | V_threshold = 1 | [None, None, 64, 300] | 0 |
| Conv1d | [None, None, 64, 300] | kernel size = 5<br>Stride = 1 | [None, 64, 300] | 20,480 |
| MultiStepIFNode | [None, 64, 300] | V_threshold = 1 | [None, None, 64, 300] | 0 |
| Dropout | [None, None, 64, 300] | – | [None, None, 64, 300] | 0 |
| Flatten | [None, None, 64, 300] | – | [None, None, 19,200] | 0 |
| SSA | [None, None, 64, 300] | Number of heads = 8 | [None, None, 19,200] | 16,960 |
| Dropout | [None, None, 19,200] | – | [None, None, 19,200] | 0 |
| MaxPool1d | [None, None, 38,400] | kernel size = 2<br>Stride = 2 | [None, 19,200] | 0 |
| Flatten | [None, 19,200] | – | [None, 19,200] | 0 |
| Linear | [None, 19,200] | – | [None, 2] | 38,400 |
| Dropout | [None, 2] | – | [None, 2] | 0 |
| MultiStepIFNode | [None, 2] | V_threshold = 1 | [None, None, 2] | 0 |
| Total parameters | – | – | – | 77,248 |

and accuracy are all between 0 and 1, and the value of the MCC is between −1 and 1. Larger values of these metrics represent better performance of the method. In addition, receiver operating characteristic (ROC) curves and the area under ROC curve (AUC) are also applied to evaluate the overall performance of recognition methods. The AUC value closer to 1 indicates better performance. Area under the precision recall curve (AUPRC) is used to evaluate the recognition performance of methods on imbalanced datasets. AUPRC calculates the area under Precision-Recall curve. The value of AUPRC indicates Pre performance at different Recall levels.

K-fold cross-validation is widely used to estimate generalization of computational methods. We validate the promoter recognition performance of our method by performing five-fold cross-validation on benchmark datasets from the seven species. Each dataset is divided into five subsets of equal size, where four subsets are used as the training datasets and the remaining subset is used as a validation dataset, alternately. The final recognition results are the average evaluation metric values of five-fold cross-validation.

# RESULTS AND DISCUSSION

## Experiments design and compared methods

In this section, we introduce our experiments design and some state-of-art deep learning methods which are used to make performance comparison with our method.

In our experiments, we first make ablation study to verify the effectiveness of our proposed feature fusion module. Then we take three-layer CSNNs and two-layer CSNNs cascaded with a multi-head SSA module as baseline methods, and compare the promoter

recognition performance of our iPro-CSAF with baseline methods on *A. thaliana* benchmark datasets.

After that, we compare our method with some advanced deep learning methods on both benchmark datasets and independent datasets. Most of these compared methods are designed only for a certain species or a certain class of species, while in our work our datasets are from seven species including both prokaryotes and eukaryotes, therefore in our comparison experiments, for different species we compare our method with different deep learning methods. For *E. coli* datasets, we compare our method with FastText N-grams (*Le et al., 2019*), iPSW(PseDNC-DL) (*Tayara, Tahir & Chong, 2020*), iPromoter-BnCNN (*Amin et al., 2020*), iPro2L-CLA (*Zhang et al., 2022*) and iProL (*Peng, Sun & Fan, 2024*). For *B. subtilis* dataset, we compare our method with CapsProm (*Moraes et al., 2022*). For the cyanobacteria dataset, the compared methods are DeePromoter (*Oubounyt et al., 2019*), CyaPromBERT (*Mai, Nguyen & Lee, 2022*) and SiamProm (*Yang et al., 2024*). For *A. thaliana* datasets, we compare our method with Depicter. For *H. sapiens* datasets, we compare our method with Depicter and DeePromClass (*Kari et al., 2023*). For *D. melanogaster* and *M. Musculus* datasets, we compare our method with Depicter, DeeProPre (*Ma et al., 2022*) and DeePromClass (*Kari et al., 2023*).

Finally, we calculate the power consumption of iPro-CSAF and compare it with that of traditional neural network with the same structure.

## Ablation study

We conduct ablation experiments to evaluate the effectiveness of feature fusion module which has a parallel structure including a convolutional spiking layer and a multi-head spiking self-attention module.

In this experiment we use *H. sapiens* non-TATA promoter datasets which are eukaryotic promoters and *E. coli* promoter datasets which are prokaryotic promoters. These two datasets are relatively more difficult to classify among the seven species. iPro-CSAF consists of two convolutional spiking layers and a spiking attention mechanism. We name the first convolution layer Conv_1 and the second convolution layer Conv_2 as shown in Fig. 1. In our feature fusion module, it contains two parallel branches and one concatenation module. The two parallel branches are Conv_2 and MSSA, respectively.

Table 4 shows the results of ablation experiments. For *H. sapiens* non-TATA promoter recognition, iPro-CSAF outperforms two-layer CSNNs (Conv_1 + Conv_2), one-layer CSNN cascaded with a MSSA (Conv_1 + MSSA) and one-layer CSNN (Conv_1) by 1.69%, 2.40% and 4.51%, respectively. For *E. coli* promoter recognition, iPro-CSAF outperforms two-layer CSNNs (Conv_1 + Conv_2), one-layer CSNN cascaded with a MSSA (Conv_1 + MSSA) and one-layer CSNN (Conv_1) by 1.95%, 2.12% and 3.11%, respectively. The results of ablation experiments show that after extracting feature with Conv_1, our feature fusion module with parallel structure can significantly improve the performance of the method and obtain higher recognition accuracy than using convolutional spiking layer or spiking attention mechanism alone to further extract features.

**Table 4 The results of ablation studies without proposed module.**

| Promoter types | Methods | Sn (%) | Sp (%) | Acc (%) | MCC | F1 |
|---|---|---|---|---|---|---|
| *H. sapiens* non-TATA | iPro-CSAF | **89.19** | **90.48** | **89.83** | **0.7968** | **0.8977** |
| | Conv_1 + Conv_2 | 87.59 | 88.68 | 88.14 | 0.7630 | 0.8808 |
| | Conv_1 + MSSA | 86.46 | 88.39 | 87.43 | 0.7488 | 0.8730 |
| | Conv_1 | 82.72 | 87.83 | 85.32 | 0.7100 | 0.8485 |
| *E.coli* promoter | iPro-CSAF | **84.81** | **89.93** | **87.37** | **0.7484** | **0.8704** |
| | Conv_1 + Conv_2 | 82.24 | 88.58 | 85.42 | 0.7098 | 0.8494 |
| | Conv_1 + SSA | 84.15 | 86.38 | 85.25 | 0.7054 | 0.8506 |
| | Conv_1 | 81.65 | 86.86 | 84.26 | 0.6862 | 0.8384 |

Note:
Bold values indicate the highest score in each column.

**Table 5 The recognition performance of iPro-CSAF and the baseline methods.**

| Promoter types | Models | Sn (%) | Sp (%) | Acc (%) | MCC | AUC |
|---|---|---|---|---|---|---|
| *A. thaliana* TATA | iPro-CSAF | **99.52** | **99.18** | **99.35** | **0.9871** | **0.9970** |
| | two-layer CSNNs + MSSA | 98.94 | 99.16 | 99.05 | 0.9810 | 0.9945 |
| | three-layer CSNNs | 99.29 | 98.91 | 99.10 | 0.9821 | 0.9952 |
| *A. thaliana* non-TATA | iPro-CSAF | **94.39** | 96.93 | **95.66** | **0.9135** | 0.9707 |
| | two-layer CSNNs + MSSA | 93.94 | 97.09 | 95.51 | 0.9107 | 0.9721 |
| | three-layer CSNNs | 93.31 | 97.13 | 95.22 | 0.9051 | 0.9673 |
| *A. thaliana* TATA&non-TATA | iPro-CSAF | 95.02 | **97.85** | **96.44** | **0.9291** | 0.9762 |
| | two-layer CSNNs + MSSA | **95.37** | 97.17 | 96.27 | 0.9256 | 0.9768 |
| | three-layer CSNNs | 95.01 | 97.17 | 96.09 | 0.9220 | 0.9740 |

Note:
Bold values indicate the highest score in each column.

## Comparison of iPro-CSAF with baseline methods

As shown in Fig. 1, our method consists of two convolutional spiking layers and a multi-head SSA module. The extracted features from the two parallel branches are fused to make decisions. In order to further evaluate the effectiveness of our method, we take three-layer CSNNs and two-layer CSNNs cascaded with a multi-head SSA module as baseline methods. We compare promoter recognition performance of iPro-CSAF with baseline methods on *A. thaliana* benchmark datasets. Table 5 shows the recognition performances of iPro-CSAF and baseline methods. We use a Mann-Whitney's U test to evaluate statistically significant difference between Acc results for *A. thaliana* TATA&non-TATA promoter recognition of different methods. Statistical difference is significant at $p$-value < 0.05. The p-values between Acc results of iPro-CSAF and three-layer CSNNs and two-layer CSNNs +MSSA both are 0.027. It is shown that the Acc results of iPro-CSAF and baseline methods are significantly different.

As shown in Table 5, iPro-CSAF can obtain better promoter recognition accuracy than baseline methods. For *A. thaliana* non-TATA promoter recognition, iPro-CSAF outperforms three-layer CSNNs by 0.44% Two-layer CSNNs cascaded with a multi-head SSA module outperforms two-layer CSNNs cascaded with a convolutional layer. This is

due to that SSA can extract long-distance dependence features and focus on the more important features in sequences. By using parallel branches of a convolutional layer and a multi-head spiking self-attention module, our iPro-CSAF outperforms two-layer CSNNs cascaded with a multi-head SSA module by 0.15%

## Comparison of iPro-CSAF with state-of-the-art methods on benchmark datasets

In this section, we compare the recognition performance of our iPro-CSAF with some state-of-the-art deep learning methods. The recognition performance of iPro-CSAF and compared methods on benchmark datasets is shown in Table 6.

First we compare our iPro-CSAF with other methods on prokaryotic promoter recognition. For the *E. coli* benchmark dataset, we compare the recognition performance of our iPro-CSAF with FastText N-grams (*Le et al., 2019*), iPSW(PseDNC-DL) (*Tayara, Tahir & Chong, 2020*), iPro2L-CLA (*Zhang et al., 2022*) and iProL (*Peng, Sun & Fan, 2024*). iPSW(PseDNC-DL) and FastText N-grams both used multiple CNN layers to make further feature extraction. Differently, iPSW(PseDNC-DL) extracted priori biological features by PseDNC, while FastText N-grams extracted priori text features by FastText natural language processing method. iPro2L-CLA used one-hot coding scheme, then adopted multiple deep learning algorithms including CNNs, LSTM, capsule networks and attention mechanism to extract promoter sequence features. iProL used longformer (*Beltagy, Peters & Cohan, 2020*), CNNs and bidirectional LSTM (BiLSTM) to extract features directly from input promoter sequences to identify *E. coli* promoters. As shown in Table 6, iPro-CSAF obviously outperforms all these compared methods in terms of Sp, Acc and MCC. The results show that our method which uses parallel combination of convolution and MSSA outperforms promoter recognition methods which used combination of CNNs, LSTM, capsule networks and attention mechanism (iPro2L-CLA, iProL), and it also outperforms CNNs which needed priori biological feature extraction (iPSW(PseDNC-DL)) or priori text feature extraction (FastText N-grams).

For *B. subtilis* promoter recognition, we compare the recognition performance of our iPro-CSAF with CapsProm. CapsProm used one-hot coding scheme and combined CNNs with capsule networks to extract features. Our iPro-CSAF outperforms CapsProm in terms of Sn, Sp, Acc, F1 and MCC. The Acc value of iPro-CSAF is higher by 2.46% than CapsProm. Moreover, the number of parameters of CapsProm is 1271127, much more than 77248 of iPro-CSAF. Therefore, our iPro-CSAF can achieve much better recognition performance than CapsProm with much lower network complexity.

For cyanobacteria promoter recognition, we compare our iPro-CSAF with DeePromoter (*Oubounyt et al., 2019*), CyaPromBERT (*Mai, Nguyen & Lee, 2022*) and SiamProm (*Yang et al., 2024*). DeePromoter was a CNNs-based promoter recognition method combined with BiLSTM. CyaPromBERT utilized Bidirectional Encoder Representations from Transformers (BERT) to perform pre-training on genomes for promoter recognition. SiamProm was a cyanobacterial promoter recognition method based on the Siamese network which contained two same subnetworks. Each subnetwork comprised four parallel modules including an embedding initializer, a k-mer attention

**Table 6 The recognition performance of iPro-CSAF and compared methods on benchmark datasets.**

| Promoter types | Predictors | Sn (%) | Sp (%) | Acc (%) | F1 | MCC | AUC | AUPR |
|---|---|---|---|---|---|---|---|---|
| *E.coli* promoter | iPro-CSAF | 84.81 | **89.93** | 87.37 | **0.8704** | **0.7484** | 0.9046 | **0.9167** |
| | iProL (*Peng, Sun & Fan, 2024*) | 84.62 | 86.61 | 85.62 | – | 0.7130 | 0.9211 | – |
| | iPro2L-CLA (*Zhang et al., 2022*) | **86.87** | 85.13 | 86.00 | – | 0.7211 | **0.9291** | – |
| | iPSW(PseDNC-DL) (*Tayara, Tahir & Chong, 2020*) | 83.34 | 86.83 | 85.10 | – | 0.7024 | 0.9250 | – |
| | FastText N-grams (*Le et al., 2019*) | 82.76 | 88.05 | 85.41 | – | 0.7090 | – | – |
| *B. subtilis* promoter | iPro-CSAF | **94.01** | **96.89** | **96.08** | **0.9299** | **0.9030** | 0.9728 | 0.9501 |
| | CapsProm (*Moraes et al., 2022*) | 89.73 | 95.05 | 93.62 | 0.8823 | 0.8393 | – | – |
| Cyanobacteria promoter | iPro-CSAF | **96.06** | 97.20 | 96.63 | 0.9661 | 0.9327 | 0.9777 | 0.9815 |
| | SiamProm (*Yang et al., 2024b*) | 95.08 | **98.56** | **96.80** | – | **0.9367** | - | - |
| | CyaPromBERT (*Mai, Nguyen & Lee, 2022*) | 94.69 | 95.58 | 95.13 | – | 0.9027 | – | – |
| | DeePromoter (*Oubounyt et al., 2019*) | 93.10 | 94.01 | 93.55 | – | 0.8711 | – | – |
| *A. thaliana* TATA | iPro-CSAF | **99.52** | **99.18** | **99.35** | **0.9936** | **0.9871** | 0.9970 | **0.9975** |
| | Depicter (*Zhu et al., 2021*) | 97.09 | 98.38 | 97.72 | 0.9776 | 0.9544 | **0.9990** | – |
| *A. thaliana* non-TATA | iPro-CSAF | 94.39 | **96.93** | 95.66 | **0.9561** | **0.9135** | 0.9707 | **0.9767** |
| | Depicter (*Zhu et al., 2021*) | **95.00** | 94.56 | 94.78 | 0.9477 | 0.8956 | **0.9830** | – |
| *A. thaliana* TATA&non-TATA | iPro-CSAF | 95.02 | **97.85** | **96.44** | **0.9638** | **0.9291** | **0.9762** | **0.9819** |
| | Depicter (*Zhu et al., 2021*) | **96.06** | 95.38 | 95.72 | 0.9570 | 0.9144 | 0.9660 | – |
| *M. musculus* TATA | iPro-CSAF | **99.53** | 99.71 | **99.62** | **0.9962** | **0.9924** | 0.9981 | **0.9986** |
| | DeePromClass (*Kari et al., 2023*) | 97.87 | 96.43 | 97.14 | 0.9716 | 0.9431 | 0.9716 | 0.9769 |
| | DeeProPre (*Ma et al., 2022*) | 99.42 | 98.26 | 98.84 | 0.9884 | 0.9768 | 0.9924 | – |
| | Depicter (*Zhu et al., 2021*) | 99.29 | 99.63 | 99.46 | 0.9947 | 0.9892 | **1** | – |
| *M. musculus* non-TATA | iPro-CSAF | 98.88 | **99.39** | **99.13** | **0.9913** | **0.9827** | 0.9937 | **0.9955** |
| | DeePromClass (*Kari et al., 2023*) | 98.25 | 98.89 | 98.57 | 0.9856 | 0.9714 | 0.9857 | 0.9901 |
| | DeeProPre (*Ma et al., 2022*) | 97.76 | 98.34 | 98.05 | 0.9805 | 0.9608 | 0.9846 | – |
| | Depicter (*Zhu et al., 2021*) | **99.20** | 97.13 | 98.15 | 0.9815 | 0.9633 | **0.9970** | – |
| *M. musculus* TATA&non-TATA | iPro-CSAF | **98.78** | 99.40 | **99.09** | **0.9909** | **0.9819** | **0.9931** | **0.9951** |
| | DeePromClass (*Kari et al., 2023*). | 98.29 | 98.95 | 98.62 | 0.9861 | 0.9723 | 0.9861 | 0.9904 |
| | DeeProPre (*Ma et al., 2022*) | 97.51 | 98.50 | 98.00 | 0.9799 | 0.9601 | 0.9851 | – |
| | Depicter (*Zhu et al., 2021*) | 98.38 | 98.17 | 98.27 | 0.9826 | 0.9654 | 0.9930 | – |
| *H. sapiens* TATA | iPro-CSAF | **98.58** | 98.53 | **98.55** | **0.9855** | **0.9710** | 0.9929 | **0.9940** |
| | DeePromClass (*Kari et al., 2023*) | 96.86 | 97.95 | 97.40 | 0.9737 | 0.9481 | 0.9740 | 0.9818 |
| | Depicter (*Zhu et al., 2021*) | 96.60 | 98.47 | 97.53 | 0.9752 | 0.9508 | **0.9940** | – |
| *H. sapiens* non-TATA | iPro-CSAF | **89.19** | **90.48** | **89.83** | **0.8977** | **0.7968** | 0.9300 | **0.9377** |
| | DeePromClass (*Kari et al., 2023*) | 85.20 | 88.95 | 86.96 | 0.8660 | 0.7404 | 0.8696 | 0.9058 |
| | Depicter (*Zhu et al., 2021*) | 88.61 | 89.14 | 88.87 | 0.8903 | 0.7774 | **0.9440** | – |
| *H. sapiens* TATA&non-TATA | iPro-CSAF | **90.00** | **90.62** | **90.31** | **0.9028** | **0.8063** | **0.9332** | **0.9405** |
| | DeePromClass (*Kari et al., 2023*) | 86.66 | 89.82 | 88.17 | 0.8791 | 0.7640 | 0.8816 | 0.9143 |
| | Depicter (*Zhu et al., 2021*) | 88.22 | 89.95 | 89.06 | 0.8922 | 0.7814 | 0.8830 | – |

(*Continued*)

| Table 6 (continued) | | | | | | | | |
|---|---|---|---|---|---|---|---|---|
| **Promoter types** | **Predictors** | **Sn (%)** | **Sp (%)** | **Acc (%)** | **F1** | **MCC** | **AUC** | **AUPR** |
| *D. melanogaster* TATA | iPro-CSAF | **98.95** | 98.33 | **98.64** | **0.9865** | **0.9729** | 0.9913 | **0.9920** |
| | DeePromClass (*Kari et al., 2023*) | 98.73 | 96.38 | 97.52 | 0.9754 | 0.9509 | 0.9753 | 0.9787 |
| | DeeProPre (*Ma et al., 2022*) | 98.01 | 94.59 | 96.41 | 0.9648 | 0.9292 | **0.9925** | – |
| | Depicter (*Zhu et al., 2021*) | 93.17 | **100** | 96.35 | 0.9647 | 0.9295 | 0.9890 | – |
| *D. melanogaster* non-TATA | iPro-CSAF | 94.08 | **95.80** | 94.94 | **0.9490** | **0.8990** | 0.9662 | **0.9717** |
| | DeePromClass (*Kari et al., 2023*) | **94.33** | 92.31 | 93.29 | 0.9337 | 0.8661 | 0.9329 | 0.9477 |
| | DeeProPre (*Ma et al., 2022*) | 93.50 | 92.91 | 92.75 | 0.9322 | 0.8644 | **0.9847** | – |
| | Depicter (*Zhu et al., 2021*) | 94.10 | 91.12 | 92.52 | 0.9220 | 0.8509 | 0.9750 | – |
| *D. melanogaster* TATA&non-TATA | iPro-CSAF | 94.14 | **96.07** | 95.11 | **0.9506** | **0.9025** | 0.9681 | **0.9729** |
| | DeePromClass (*Kari et al., 2023*) | **94.48** | 94.00 | 94.24 | 0.9425 | 0.8848 | 0.9424 | 0.9563 |
| | DeeProPre (*Ma et al., 2022*) | 93.08 | 93.59 | 93.35 | 0.9333 | 0.8675 | **0.9852** | – |
| | Depicter (*Zhu et al., 2021*) | 92.09 | 91.42 | 91.74 | 0.9149 | 0.8348 | 0.9750 | – |

**Note:**
 Bold values indicate the highest score in each column.

module, a bi-directional context catcher and a nearest-neighbor aggregator. It was a complex deep learning method. Our iPro-CSAF outperforms DeePromoter and CyaPromBERT in terms of Sn, Sp, Acc and MCC. The Acc value of iPro-CSAF is 96.63%, slightly lower than 96.80% of SiamProm. But parameter numbers of SiamProm is 7,253,376, much more than 77,248 of iPro-CSAF. This shows that our method can achieve comparable accuracy with SiamProm, while has much lower network complexity than SiamProm.

We then compare our iPro-CSAF with other methods on eukaryotic promoter recognition. For the *A. thaliana* promoter recognition, we compare the recognition performance of iPro-CSAF with Depicter (*Zhu et al., 2021*). Depicter first used one-hot coding, then adopted CNNs and capsule networks to extract promoter features. For all the three recognition tasks of *A. thaliana*, our iPro-CSAF outperforms Depicter in terms of Sp, Acc, F1 and MCC. The Acc value of iPro-CSAF is higher by 1.63% on *A. thaliana* TATA promoter recognition. It shows our parallel combination of convolution and MSSA is superior to serial combination of convolution and capsule networks (Depicter).

For *M. musculus* promoter recognition, we compare iPro-CSAF with DeePromClass (*Kari et al., 2023*), DeeProPre (*Ma et al., 2022*) and Depicter. DeePromClass first used one-hot coding, then adopted CNNs and LSTM networks to extract promoter features. DeeProPre first encoded promoter sequences by a word embedding layer, then extracted further features by combing BiLSTM with CNNs. For all the three recognition tasks of this species, iPro-CSAF outperforms all the three compared methods in terms of Sp, Acc, F1, MCC and AUPR, and the Acc values of iPro-CSAF are all above 99% on *M. musculus* promoter recognition. It shows our parallel combination of convolution and MSSA is also superior to serial combination of convolution and LSTM networks.

For *H. sapiens* promoter recognition, we compare iPro-CSAF with DeePromClass (*Kari et al., 2023*) and Depicter. For all the three recognition tasks of *H. sapiens*, our iPro-CSAF

outperforms Depicter in terms of Sn, Sp, Acc, F1, MCC and AUPR. For the TATA&non-TATA promoter recognition, the Acc of iPro-CSAF is 1.25% and 2.14% higher than Depicter and DeePromClass, respectively.

For *D. melanogaster* promoter recognition, the compared methods are DeePromClass (*Kari et al., 2023*), DeeProPre (*Ma et al., 2022*) and Depicter, the same as *M. musculus* promoter recognition. iPro-CSAF outperforms all the three methods in terms of Acc, F1, MCC and AUPR. The Acc values of iPro-CSAF are higher by 2.29%, 2.42% and 3.37% than Depicter on the three types of *D. melanogaster* promoter recognition, respectively. We also compare the network complexity of our method with Depicter, DeeProPre and DeePromClass. And the parameter numbers of Depicter, DeeProPre and DeePromClass are 4,177,744, 606,337 and 38,329,410 respectively, which are much more than 77,248 of iPro-CSAF. It shows that our iPro-CSAF can achieve better recognition performance than these methods with much lower network complexity.

From our results on prokaryotic promoter recognition, it can be seen that our method which uses parallel combination of convolution and MSSA outperforms promoter recognition methods which used combination of CNNs, LSTM, capsule networks and attention mechanism (iPro2L-CLA, iProL, CapsProm, DeePromoter), and it also outperforms CNNs which needed priori biological feature extraction (iPSW(PseDNC-DL) or priori text feature extraction (FastText N-grams). Our iPro-CSAF can achieve comparable accuracy with promoter recognition method which used combination of text features and biological sequence features (SiamProm), while has much lower network complexity. For *B. subtilis* promoter recognition, the Acc value of iPro-CSAF is higher than CapsProm. It indicates good performance of our method on unbalanced datasets. From our results on eukaryotic promoter recognition, it can be seen that our method outperforms promoter recognition methods which used combination of CNNs and LSTM (DeePromClass and DeeProPre), and it also outperforms method which used combination of CNNs and capsule networks (Depicter).

As a whole, the Acc values of iPro-CSAF on eukaryotic promoter recognition are higher than that on prokaryotic promoter recognition. The results show that our iPro-CSAF with simple structure has good ability and generalization to extract promoter features.

## Comparison of iPro-CSAF with state-of-the-art methods on independent datasets

The independent datasets constructed in 'Materials and Methods' are used to validate the promoter recognition performance of our proposed method. The recognition performance of iPro-CSAF and compared methods on independent datasets is shown in Table 7. First we compare our iPro-CSAF with other methods on prokaryotic promoter recognition. For *E. coli* promoter recognition, our iPro-CSAF is compared with iProL (*Peng, Sun & Fan, 2024*), iPro2L-CLA (*Zhang et al., 2022*), iPSW(PseDNC-DL) (*Tayara, Tahir & Chong, 2020*) and iPromoter-BnCNN (*Amin et al., 2020*). The first three methods are also used as compared methods for *E. coli* promoter recognition on benchmark dataset. iPromoter-BnCNN used four parallel three-layer CNNs to extract sequence features and each of the four CNNs used different encoding methods. As shown in Table 7, iPro-CSAF

**Table 7 The recognition performance of iPro-CSAF and compared methods on independent datasets.**

| Promoter types | Predictors | Sn (%) | Sp (%) | Acc (%) | F1 | MCC | AUC | AUPR |
|---|---|---|---|---|---|---|---|---|
| *E. coli* promoter | iPro-CSFA | **97.24** | **90.59** | **93.92** | **0.9411** | **0.8803** | 0.9532 | **0.9565** |
| | iProL (*Peng, Sun & Fan, 2024*) | 95.80 | 89.80 | 92.80 | 0.9300 | 0.8580 | **0.9780** | – |
| | iPro2L-CLA (*Zhang et al., 2022*) | 86.27 | 84.80 | 85.53 | – | 0.7114 | – | – |
| | iPromoter-BnCNN (*Amin et al., 2020*) | 88.30 | 88.00 | 88.20 | – | 0.7630 | – | – |
| | iPSW(PseDNC-DL) (*Tayara, Tahir & Chong, 2020*) | 89.72 | 81.61 | 85.56 | – | 0.7156 | – | – |
| *A. thaliana* TATA | iPro-CSFA | **99.37** | 99.21 | 99.29 | **0.9929** | 0.9858 | 0.9967 | 0.9970 |
| | Depicter (*Zhu et al., 2021*) | 98.43 | **99.36** | **99.37** | 0.9889 | **0.9889** | **0.9990** | – |
| *A. thaliana* non-TATA | iPro-CSFA | 94.51 | **97.16** | 95.84 | **0.9578** | **0.9171** | 0.9693 | **0.9786** |
| | Depicter (*Zhu et al., 2021*) | **96.45** | 94.27 | 95.33 | 0.9528 | 0.9069 | **0.9830** | – |
| *A. thaliana* TATA&non-TATA | iPro-CSFA | 95.00 | **97.93** | 96.46 | **0.9641** | **0.9296** | 0.9761 | **0.9816** |
| | Depicter (*Zhu et al., 2021*) | **96.27** | 95.44 | 95.85 | 0.9583 | 0.9171 | **0.9860** | – |
| *M. musculus* TATA | iPro-CSFA | **100** | 99.67 | 99.84 | **0.9984** | **0.9968** | 0.9999 | **0.9999** |
| | DeePromClass (*Kari et al., 2023*). | 97.41 | 97.72 | 97.56 | 0.9756 | 0.9513 | 0.9756 | 0.9821 |
| | DeeProPre (*Ma et al., 2022*) | 99.61 | **99.68** | 99.65 | 0.9964 | 0.9929 | 0.9999 | – |
| | Depicter (*Zhu et al., 2021*) | 99.35 | 99.35 | 99.35 | 0.9935 | 0.9870 | 1 | – |
| *M. musculus* non-TATA | iPro-CSFA | 98.62 | **99.48** | **99.05** | **0.9905** | **0.9810** | 0.9937 | **0.9955** |
| | DeePromClass (*Kari et al., 2023*). | **98.86** | 98.86 | 98.86 | 0.9886 | 0.9772 | 0.9886 | 0.9914 |
| | DeeProPre (*Ma et al., 2022*) | 97.99 | 98.17 | 98.08 | 0.9808 | 0.9617 | **0.9970** | – |
| | Depicter (*Zhu et al., 2021*) | 98.75 | 97.83 | 98.29 | 0.9828 | 0.9658 | 0.9970 | – |
| *M. musculus* TATA&non-TATA | iPro-CSFA | 98.76 | **99.59** | **99.17** | **0.9917** | **0.9835** | 0.9954 | **0.9970** |
| | DeePromClass (*Kari et al., 2023*). | **98.88** | 98.92 | 98.90 | 0.9890 | 0.9780 | 0.9890 | 0.9918 |
| | DeeProPre (*Ma et al., 2022*) | 98.19 | 98.09 | 98.14 | 0.9814 | 0.9629 | **0.9960** | – |
| | Depicter (*Zhu et al., 2021*) | 98.13 | 97.73 | 97.93 | 0.9792 | 0.9585 | 0.9930 | – |
| *H. sapiens* TATA | iPro-CSFA | **97.61** | **98.98** | **98.29** | **0.9828** | **0.9660** | 0.9922 | **0.9937** |
| | DeePromClass (*Kari et al., 2023*). | 95.07 | 98.58 | 96.75 | 0.9670 | 0.9358 | 0.9676 | 0.9801 |
| | Depicter (*Zhu et al., 2021*) | 96.96 | 97.93 | 97.44 | 0.9745 | 0.9489 | **0.9940** | – |
| *H. sapiens* non-TATA | iPro-CSFA | 87.71 | **91.59** | 89.65 | **0.8945** | **0.7936** | 0.9353 | **0.9424** |
| | DeePromClass (*Kari et al., 2023*). | 87.03 | 90.28 | 88.59 | 0.8835 | 0.7725 | 0.8859 | 0.9176 |
| | Depicter (*Zhu et al., 2021*) | **88.26** | 89.08 | 88.67 | 0.8873 | 0.7734 | **0.9440** | – |
| *H. sapiens* TATA&non-TATA | iPro-CSFA | **87.81** | **91.93** | **89.87** | **0.8966** | **0.7981** | **0.9354** | **0.9418** |
| | DeePromClass (*Kari et al., 2023*). | 87.28 | 91.18 | 89.13 | 0.8886 | 0.7836 | 0.8913 | 0.9225 |
| | Depicter (*Zhu et al., 2021*) | 87.57 | 90.91 | 89.17 | 0.8939 | 0.7841 | 0.8830 | – |
| *D. melanogaster* TATA | iPro-CSFA | **99.23** | **97.30** | 98.26 | **0.9828** | **0.9654** | 0.9898 | **0.9906** |
| | DeePromClass (*Kari et al., 2023*). | 98.80 | 95.88 | 97.30 | 0.9734 | 0.9464 | 0.9730 | 0.9765 |
| | DeeProPre (*Ma et al., 2022*) | 98.46 | 94.98 | 96.73 | 0.9677 | 0.9350 | **0.9930** | – |
| | Depicter (*Zhu et al., 2021*) | 92.11 | 91.21 | 95.37 | 0.9554 | 0.9101 | 0.9890 | – |
| *D. melanogaster* non-TATA | iPro-CSFA | 96.01 | 95.58 | **95.80** | **0.9581** | **0.9160** | 0.9667 | **0.9705** |
| | DeePromClass (*Kari et al., 2023*). | 95.47 | 93.32 | 94.37 | 0.9444 | 0.8877 | 0.9437 | 0.9556 |
| | DeeProPre (*Ma et al., 2022*) | **96.07** | 91.41 | 93.74 | 0.9388 | 0.8759 | **0.9850** | – |
| | Depicter (*Zhu et al., 2021*) | 94.01 | **97.83** | 91.50 | 0.9212 | 0.8511 | 0.9750 | – |

| Table 7 (continued) | | | | | | | | |
|---|---|---|---|---|---|---|---|---|
| Promoter types | Predictors | Sn (%) | Sp (%) | Acc (%) | F1 | MCC | AUC | AUPR |
| *D. melanogaster* TATA&non-TATA | iPro-CSFA | 94.59 | **96.03** | **95.31** | **0.9528** | **0.9063** | 0.9685 | **0.9740** |
| | DeePromClass (*Kari et al., 2023*). | **96.30** | 92.56 | 94.35 | 0.9446 | 0.8877 | 0.9435 | 0.9539 |
| | DeeProPre (*Ma et al., 2022*) | 93.97 | 93.92 | 93.97 | 0.9397 | 0.8799 | **0.9850** | – |
| | Depicter (*Zhu et al., 2021*) | 92.13 | 92.23 | 92.18 | 0.9219 | 0.8437 | 0.9750 | – |

**Note:**
Bold values indicate the highest score in each column.

outperforms all the compared methods in terms of Sn, Sp, Acc and MCC. The Acc value of iPro-CSAF is significantly higher than iProL, iPro2L-CLA, iPromoter-BnCNN and iPSW (PseDNC-DL) by 1.12%, 8.39%, 5.72% and 8.36%, respectively. The results again show that our method outperforms promoter recognition methods which used combination of CNNs, LSTM, capsule networks and attention mechanism (iPro2L-CLA), CNNs which needed priori biological feature extraction (iPSW(PseDNC-DL)). And it also outperforms method which used parallel three-layer CNNs (iPromoter-BnCNN).

Then we compare our iPro-CSAF with other methods on eukaryotic promoter recognition. For the *A. thaliana* promoter recognition, we compare iPro-CSAF with Depicter (*Zhu et al., 2021*) which is also the compared method on benchmark datasets of *A. thaliana*. For all the three types of promoter recognition, iPro-CSAF outperforms Depicter in terms of F1. And for the non-TATA promoter recognition and TATA&non-TATA promoter recognition, the Acc values of iPro-CSAF are slightly higher than that of Depicter. For the TATA promoter recognition, the Acc value of iPro-CSAF is 99.29%, very close to that of 99.37% of Depicter. But the parameter numbers of Depicter is 4,177,744, much more than 77,248 of iPro-CSAF.

For the *M. musculus* promoter recognition, we compare iPro-CSAF with DeePromClass (*Kari et al., 2023*), DeeProPre (*Ma et al., 2022*) and Depicter, which are also the compared methods on benchmark datasets. For all the three types of promoter recognition, iPro-CSAF outperforms all the compared methods in terms of F1, Acc, MCC and AUPR.

For *H. sapiens* promoter recognition, we compare iPro-CSAF with DeePromClass (*Kari et al., 2023*) and Depicter, which are also the compared methods on benchmark datasets. For all the three types of promoter recognition, iPro-CSAF outperforms all the compared methods in terms of Sp, F1, Acc, MCC and AUPR. The AUC value of iPro-CSAF is higher than Depicter by 5.24% on TATA&non-TATA promoter recognition.

For *D. melanogaster* promoter recognition, we compare iPro-CSAF with DeePromClass (*Kari et al., 2023*), DeeProPre (*Ma et al., 2022*) and Depicter, the same as benchmark datasets. iPro-CSAF outperforms all the three compared methods in terms of Acc, F1, MCC and AUPR. The Acc values of iPro-CSAF is higher than Depicter by 2.89%, 4.3% and 3.13%, respectively, on the three types of *D. melanogaster* promoter recognition.

The results on prokaryotic promoter recognition in Table 7 again show that our method outperforms promoter recognition methods which used combination of CNNs, LSTM, capsule networks and attention mechanism (iPro2L-CLA), CNNs which needed priori biological feature extraction (iPSW(PseDNC-DL)). And it also outperforms methods

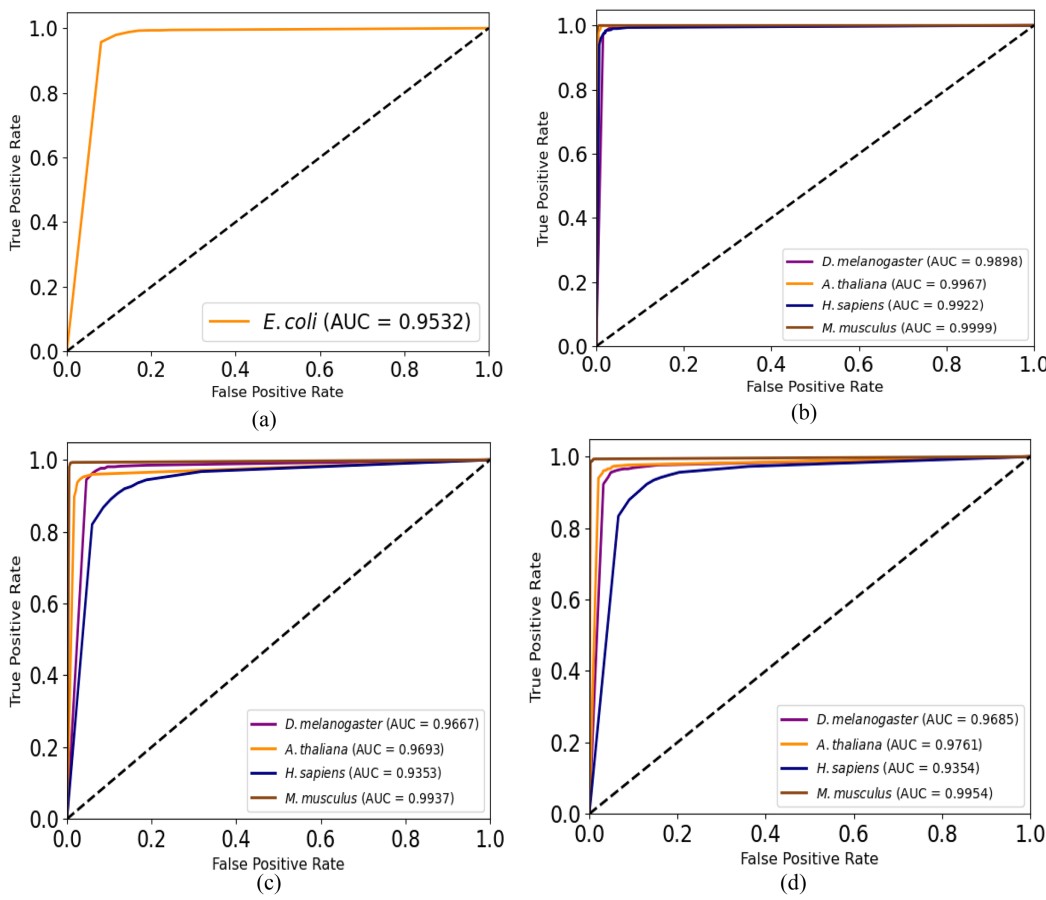

**Figure 3 ROC curves for promoter recognition on independent datasets.** (A) *E. coli* promoters. (B) TATA promoters. (C) non-TATA promoters. (D) TATA & non-TATA promoters.

which used parallel three-layer CNNs (iPromoter-BnCNN). The results on eukaryotic promoter recognition in Table 7 again show our method is superior to serial combination of convolution and LSTM networks (DeePromClass and DeeProPre), and also superior to combination of convolution and capsule networks (Depicter). Our method shows good ability to identify promoters from multiple species and good generalization for promoter recognition.

Moreover, we plot ROC curves of our model for promoter recognition on independent datasets as shown in Fig. 3. All the curves cover almost entire area, and the values of AUC are close to 1, which indicates the good ability of predicting promoters.

## The power consumption of iPro-CSAF

To evaluate the power consumption of our method, we calculate the power consumption using Zhou's method (*Zhou et al., 2022*). We calculate the power consumption of iPro-CSAF for *A. thaliana* TATA&non-TATA promoter recognition and compare it with that of traditional neural network with the same structure.

**Table 8 Power consumption of iPro-CSAF on *A. thaliana* TATA & non-TATA benchmark dataset.**

| Models | Power consumption (mJ) | |
|---|---|---|
| | Without spiking neurons | With spiking neurons |
| iPro-CSAF | 11.38 | 2.69 |

To calculate the power consumption of spiking neural networks, we first need to calculate the number of spike-based accumulate (AC) operations SOPs (*Merolla et al., 2014*) which is represented as follows:

$$SOPs(l) = f_r \times T \times FLOPs(l) \tag{15}$$

where l is a block or layer in SNNs. $f_r$ refers to firing rate of spiking neurons of this block or layer in SNNs. T is the simulation time steps. FLOPs(l) refers to the number of multiply-and-accumulate (MAC) operations of l in traditional neural networks. It is assumed that the MAC and AC operations are implemented on the 45 nm hardware, where the power consumption of a MAC operation is $E_{MAC} = 4.6$ pJ and the power consumption of an AC operation is $E_{AC} = 0.9$ pJ. The power consumption calculation equation of iPro-CSAF is as follows:

$$P_{iPro-CSAF} = E_{MAC} \cdot FLOPs(Conv\_1) + E_{AC} \cdot (SOPs(Conv\_2 + MSSA) + SOPs(FC)) \tag{16}$$

where $E_{MAC} \cdot FLOPs(Conv\_1)$ is the power consumption calculation of Conv_1. It performs traditional convolutional operation and there is no spike in this layer. $E_{AC} \cdot (SOPs(Conv\_2 + MSSA) + SOPs(FC))$ is power consumption calculation of Conv_2, MSSA and the fully connected layer FC. The feature maps generated by Conv_1 are encoded into spikes train. Therefore, they perform spike-based operations. And we calculate the power consumption of traditional neural network with the same structure as iPro-CSAF for comparison. Its power consumption equation is as follows:

$$P_{ANN} = E_{MAC} \cdot (FLOPs(Conv\_1) + FLOPs(Conv\_2 + MSSA) + FLOPs(FC)) \tag{17}$$

Table 8 shows the power consumption of iPro-CSAF and that of traditional neural network with the same structure of iPro-CSAF on *A. thaliana* TATA&non-TATA benchmark dataset. As shown in Table 8, the power consumption of spiking neural network with spiking neurons is significantly lower than traditional neural network with the same structure. It indicates that our SNN method can achieve good promoter recognition results with low power consumption.

## CONCLUSIONS

In this article, we propose a spiking-based promoter recognition model named iPro-CSAF which combines convolutional spiking neural networks with multi-head spiking self-attention mechanism in a parallel structure to extract the spatio-temporal features of promoter sequences. The results of ablation study and comparison study on multiple species show that our method which uses parallel combination of convolution and MSSA

can effectively extract promoter features of both prokaryotes and eukaryotes. Comparison results show that our method outperforms methods which used parallel CNN layers, methods which combined CNNs with capsule networks, attention mechanism, LSTM or BiLSTM, and CNNs-based methods which needed priori biological or text feature extraction. And it can achieve better performance than some transformer-based methods. Moreover, our method has much fewer network parameters than some compared promoter recognition methods. We also prove that the power consumption of our iPro-CSAF with spiking neurons is significantly lower than traditional neural network with the same structure. Our method shows good ability to identify promoters from multiple species and good generalization for promoter recognition. Our work proves that spiking neural networks have advantages in extracting promoter sequence features to identify promoters.

There are also some areas in which our method can be improved. According to our results, generally our recognition accuracy on prokaryotic promoters is lower than that on eukaryotic promoters. In our experiments, prokaryotic promoters are shorter (length 81 bp) than eukaryotic promoters (length 300 bp). Features are more difficult to be learned by our networks from short input sequences than from long input sequences. Therefore, prokaryotic promoters are more difficult to be recognized than eukaryotic promoters. In our future work, we will improve the ability of our model to extract features of short promoter sequences. Multi-channel deep spiking neural network ensemble method might be adopted by learning and fusing multi-view features of short promoter sequences. In addition, it has been observed that biological neurons have adaptive membrane potential threshold rather than fixed threshold, and adaptive spike threshold enables robust and temporally precise neuronal encoding which contributes to the fine tune of networks. Therefore, adaptive adjustment of membrane potential threshold of spiking neurons will be added to improve the feature extraction ability of our SNNs in our future work.

### Funding
This work was supported by the National Natural Science Foundation of China (No. 31801104). The funders had no role in study design, data collection and analysis, decision to publish, or preparation of the manuscript.

### Grant Disclosures
The following grant information was disclosed by the authors:
 National Natural Science Foundation of China: 31801104.

### Competing Interests
The authors declare that they have no competing interests.

### Author Contributions
• Qian Zhou conceived and designed the experiments, authored or reviewed drafts of the article, and approved the final draft.

- Jie Meng conceived and designed the experiments, performed the experiments, analyzed the data, performed the computation work, prepared figures and/or tables, and approved the final draft.
- Hao Luo analyzed the data, authored or reviewed drafts of the article, and approved the final draft.

## Data Availability

The data sets are available at Zenodo: Zhou, Q. (2024). Datasets used for iPro-CSAF [Data set]. Zenodo. https://doi.org/10.5281/zenodo.13999541.

The datasets and codes are available in the Supplemental Files.

## Supplemental Information

Supplemental information for this article can be found online at http://dx.doi.org/10.7717/peerj-cs.2761#supplemental-information.

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
