# Peer review of "iPro-CSAF: identification of promoters based on convolutional spiking neural networks and spiking attention mechanism"

_PeerJ Computer Science, doi:10.7717/peerj-cs.2761_

## Round 0.1 · original submission · Major Revisions

In addition to many absolutely MUST DO comments of the reviewers I would propose to do some changes as well. I understand that the manuscript is submitted to PeerJ Computer Science, and the authors are not obliged to demonstrate their knowledge in biological issues. However, I would recommend to expand biological explanations and make them more accurate. Differences between prokaryotic and eukaryotic promoters must be clearly stated everywhere: in Introduction, Data and Methods, Results and Conclusions. Moreover, I would propose that one or more experts in promoters check the text to make necessary biological corrections. These experts may be very useful in selection of proper datasets as well.

Reviewer 1 ·

Basic reporting

This paper proposed a convolutional spiking neural network combined with spiking attention mechanism for promoter recognition. The topic is an interesting and useful work in its field, but there are still many problems that need to be modified.

Major comments:
1. The introduction to the spiking neural model in the methods section actually belongs to the background of related research and is not an innovation of this paper.
2. The description of the selected comparison methods in the experimental section is unclear. Since this paper uses multiple comparison methods and datasets, it is recommended to add a section before presenting the experimental results to briefly introduce the experimental design and comparison methods. And why do the authors compare with different methods on different species? Can they compare with same methods?
3. The paper frequently mentions iPro-CSAF as a model with low complexity. However, there is a lack of a clear explanation of this characteristic in the methods section.
4. In the comparison experiments, the performance of SiamProm is superior to iPro-CSAF, but the explanation in the text is that "our model is a much lower complexity network than SiamProm.". This explanation does not analyze the issue in depth and lacks persuasiveness.
5. It is recommended to provide a brief introduction to the evaluation metrics or calculation methods for power consumption. Additionally, why was the A.thaliana TATA/non-TATA promoter recognition dataset chosen for power consumption validation?
6. The Conclusion section largely reiterates the experimental results. It is suggested to add analysis and discussion on future directions for the model.
7. The paper lacks a detailed discussion of iPro-CSAF's innovative aspects.

Minor comments:
1. It is recommended to change "iPro-CSAF is an effectiveness computational model" in the last sentence of the abstract to "iPro-CSAF is an effective computational model."
2. In the Introduction section, line 52, "GpG" should be corrected to "CpG."
3. In line 242, the sentence "Zhou first proposed spike-form attention calculation methods." should be deleted, as this content has already been mentioned earlier and does not need to be repeated.
4. The terms "spiking convolutional layers" and "convolutional spiking layers" are used in the text. Please verify if they represent the same concept. If so, it is recommended to unify the expression.
5. Table 4 and Table 5 already present comparison results in detail. Figure 3 and Figure 4 each display part of the experimental results from the preceding tables. Is it necessary to include these figures?

Experimental design

no comment

Validity of the findings

no comment

Reviewer 2 ·

Basic reporting

1.English expressions need to be edited more careful and more native, in this manuscript, there are some mistakes.
2. The workflow in Figure 1 is unspecific and less organized. It's more like a stack of terminologies than a higher-level summary of the existing data and methodologies, please improve it.
3. I suggest the authors should elaborate their motivation in the manuscript. This method seems to be borrowed from a different application and directly applied here to a new subject. What is the novelty and technicality of their work?
4. Advancements in computational capabilities have led to the widespread application of machine learning techniques, especially deep learning, across various bioinformatics fields. Important computational models in these fields should be cited.
5. The authors should carefully check and unify the information of references. Some references lack the information of volume or contain the wrong page number, such as on lines 474 and 521.
6. Future work and limitations of the proposed algorithm should be addressed detailly in the manuscript for further research.
7. The authors should have a more comprehensive comparison and consider adding more state-of-the-art models into evaluation comparison.
8. In fact, many other conventional evaluation metrics (e.g. F1 and AUPR) should be provided to further evaluate the performance between their model and other methods.
9. Ablation study is missing. If the overall model performance is reduced with the removal of a module, this module is convincingly supported for its supportive contribution to the overall model.

Experimental design

n/a

Validity of the findings

n/a

Additional comments

n/a

---

## Round 0.2 · accepted · Accept

I am happy that the authors compromised all critical comments of the both reviewers.

Reviewer 1 ·

Basic reporting

no comment

Experimental design

no comment

Validity of the findings

no comment

Additional comments

This paper can be accepted since the authors have revised it according to the comments of the reviewers.

Reviewer 2 ·

Basic reporting

n/a

Experimental design

n/a

Validity of the findings

n/a

Additional comments

n/a